# Effects of ploidy and salmonid alphavirus infection on the skin and gill microbiome of Atlantic salmon (*Salmo salar*)

Ryan Brown[1], Lindsey Moore[2¤], Amir Mani [1], Sonal Patel[2,3]*, Irene Salinas [1]*

**1** Department of Biology, Center for Evolutionary and Theoretical Immunology (CETI), The University of New Mexico, Albuquerque, New Mexico, United States of America, **2** Institute of Marine Research, Nordnes, Bergen, Norway, **3** Norwegian Veterinary Institute, Bergen, Norway

¤ Current address: Department of Biological Sciences, University of Bergen, Bergen, Norway
* sonal.patel@hi.no (SP); isalinas@unm.edu (IS)

**Data Availability Statement:** All data is available at NCBI BioProject accession number PRJNA565540.

**Funding:** JFB was funded by the UNM PREP Program funded by the National Insitute of Health. The Norwegian Research Council (224885/E40a) to

## Abstract

The microbial communities that live in symbiosis with the mucosal surfaces of animals provide the host with defense strategies against pathogens. These microbial communities are largely shaped by the environment and the host genetics. Triploid Atlantic salmon (*Salmo salar*) are being considered for aquaculture as they are reproductively sterile and thus cannot contaminate the natural gene pool. It has not been previously investigated how the microbiome of triploid salmon compares to that of their diploid counterparts. In this study, we compare the steady-state skin and gill microbiome of both diploid and triploid salmon, and determine the effects of salmonid alphavirus 3 experimental infection on their microbial composition. Our results show limited differences in the skin-associated microbiome between triploid and diploid salmon, irrespective of infection. In the gills, we observed a high incidence of the bacterial pathogen *Candidatus Branchiomonas*, with higher abundance in diploid compared to triploid control fish. Diploid salmon infected with SAV3 showed greater histopathological signs of epitheliocystis compared to controls, a phenomenon not observed in triploid fish. Our results indicate that ploidy can affect the alpha diversity of the gills but not the skin-associated microbial community. Importantly, during a natural outbreak of *Branchiomonas* sp. the gill microbiome of diploid Atlantic salmon became significantly more dominated by this pathogen than in triploid animals. Thus, our results suggest that ploidy may play a role on Atlantic salmon gill health and provide insights into co-infection with SAV3 and *C. Branchiomonas* in Atlantic salmon.

## Introduction

Atlantic salmon (*Salmo salar L.*) are among the most widely farmed finfish species globally, with annual production exceeding two million tons. The sustainability of the salmon farming industry must be closely monitored due to numerous environmental and health concerns. One environmental concern is the risk of escaped salmon interfering with the natural gene

SP. The funders had no role in study design, data collection and analysis, decision to publish or preparation of the manuscript.

**Competing interests:** The authors have declared that no competing interests exist.

pool [1]. In order to resolve this issue, the use of triploid salmon has gained popularity, as they are reproductively sterile. Triploid Atlantic salmon exhibit some physiological differences compared to diploid fish, including metabolic deficiencies, which contribute to a high incidence of cataracts and skeletal abnormalities [2, 3]. These can be prevented by dietary supplements and rearing in cooler water temperatures [4, 5], which makes Norway an ideal location for production of triploid salmon. Additionally, triploid Atlantic salmon have been shown to display a less robust B-cell response following vaccination compared to diploids[6], suggesting ploidy may affect disease resistance. Similarly, triploid Chinook salmon *(Oncorhynchus tshawytscha)* display higher mortality six days following challenge with *Vibrio anguillarum*, as well as decreased IgM and MHC-II expression compared to diploids [7].

Economic losses due to disease outbreaks are one of the major concerns for the aquaculture sector worldwide. Atlantic salmon are susceptible to parasitic, bacterial, fungal and viral infections [8–11]. One of the more prominent viral infections affecting salmon farming is Pancreas disease (PD) caused by salmonid alphavirus (SAV). Necrosis of internal organs, particularly the heart and pancreas, as well as significant weight loss are typical signs of SAV infections. Six different SAV subtypes (SAV1-6) have been determined based on phylogenetic analysis[12]. Of these, SAV2 and SAV3 have been identified as the isolates causing PD in Norway, with SAV3 presenting the highest mortality rates[13, 14]. SAV3 outbreaks typically occur in the seawater phase of the Atlantic salmon lifecycle, with an increased susceptibility reported in earlier post-transfer stages [15]. The transition from freshwater to seawater, otherwise known as smoltification, is a fundamental biological process in the salmon life cycle that requires dramatic physiological adaptations[16], including shifts in various components of the immune system [17, 18], gill osmoregulatory physiology, hormonal changes and the reshaping of the skin-associated microbiome [19].

The microbiome plays an important role in several host defense mechanisms, including antiviral defense [20]. Commensal microorganisms are found in association with every mucosal surface. These surfaces are continuously exposed to pathogens in the external environment and provide the first line of defense against pathogen invasion. Experimental infection of Atlantic salmon with SAV3 causes dysbiosis in the skin microbiome of diploid Atlantic salmon [21] in a time and dose dependent manner. Importantly, triploid salmon have been reported to accumulate SAV3 prevalence at a lower rate than diploid salmon [22], and this observation may be attributed to differences in their microbiome. Thus, the goals of this study were first to investigate whether diploid and triploid Atlantic salmon harbor different microbial communities associated with their gills and skin. Second, using a SAV3 bath challenge model, we aimed to evaluate the changes in the gill and skin microbiome of diploid and triploid salmon following infection with SAV. Our results indicate that ploidy can affect the alpha diversity of the gills but not the skin-associated microbial community. Importantly, during a natural outbreak of *Branchiomonas* sp. the gill microbiome of diploid Atlantic salmon became significantly more dominated by this pathogen than in triploid animals. Thus, our results suggest that ploidy may play a role on Atlantic salmon gill health.

## Materials and methods

### Animals

AquaGen AS supplied diploid and triploid Atlantic salmon eyed eggs produced from same batch of fertilization. Triploidisation was performed according to Johnstone and Stet [23]. In March 2015, Atlantic salmon eyed eggs both diploid and triploid (strain, AquaGen® Atlantic QTL-innOva® IPN/PD) were purchased from AquaGen AS and transferred to Matre Research Station, Institute of Marine Research, Norway. The eyed eggs were incubated at 6˚C,

until hatching. The diploid and triploid fry were kept in separate tanks with same water source. The salmon fry were first fed with commercial feed, with no extra nutrient additions for triploids (Skretting AS). The photoperiod was kept at L:D 24:0 until November 2015, when it was changed to L:D 12:12. On the 15th January 2016, the photoperiod was switched back to L:D 24:0 to induce parr-smolt transformation. In mid-February 2016, diploid and triploid Atlantic salmon smolts were transferred to the experimental facilities at the Industrial and Aquatic Laboratory (ILAB), Bergen, Norway. Before transport, 10 triploid and diploid fish were euthanized and gill and heart tissue were sampled and placed in 1 mL RNAlater (Thermo Fisher). Fish were maintained and transported in freshwater throughout the initial period and transferred to seawater 4 days prior to challenge when the parr-smolt transformation was complete. All fish were kept at 12°C, fed according to appetite with commercial feed, containing no extra nutrient additions for triploids (Skretting AS), and starved for 24 hours before sampling and handling. Fish were anaesthetized using 10 mg/L Metomidate and 60 mg/L Benzocaine before handling and euthanized with 10 mg/L Metomidate and 160 mg/L of Benzocaine before sampling.

## Experimental SAV infection

We utilized a bath challenge model where SAV3 shed by shedder fish was used to challenge naïve diploid and triploid salmon following the procedure published earlier [15, 22]. Diploid and triploid smolts were transferred to seawater (30 ‰) 4 days prior to the start of the experiment and maintained at 12°C. The day the experiment started the water flow was stopped for one hour in the shedder tanks and at the end of this hour, the shedder fish were removed and euthanized. The tank water from the shedder tanks was pooled, the sampled to quantify viral load [22] and then diluted 1:4 in uncontaminated water. One hundred liters of the diluted seawater containing infectious virus (shed by the shedder fish) was added into each of the four 150 L tanks in which the fish were exposed to SAV3. Two of these four tanks containing SAV3 were populated with 55 diploid fish per tank and the other two tanks with 55 triploid fish per tank. Four tanks containing the same volume of water (100 L) without SAV3 added were populated with 55 diploid or triploid fish as non-infected control groups. The exposure to SAV3 was carried out for 6 hours, before the flow was re-started in all tanks as published earlier [15]. The exposed fish were set to 14°C and maintained at this temperature for experimental period of 21 days.

Samples of water collected from the shedder tanks contained an average of 160 copies of SAV3 RNA per liter and the TCID50 assay revealed 48 TCID50 SAV3 per liter. The 1:4 diluted shedder water used for exposure to potentially infect the diploid and triploid salmon had so low virus amounts that it did not provide reliable results in the copy number analysis and, therefore, the result for the undiluted water was used and divided by 4 (the dilution factor).

This study was carried out in strict accordance with the Care and Use of Laboratory Animals recommended by the Food and Safety authorities in Norway. The protocol was approved by Norwegian Animal Research Authority (Approval ID: 8413).

The fish in experiment were monitored daily by trained fish health specialists to make sure that the ethical aspects were secured. There was no mortality and none of the fish had disease symptoms or showed signs of stress which were the set criteria to euthanize them and remove from the trial to comply with humane endpoint.

## Sampling

Samples were collected for testing viral status (before and post SAV infection), microbiome analyses and histology as described below. To check the health status of the fish before

transport to the challenge facility, heart and gill samples (N = 10) were taken and stored in RNAlater at the rearing facility the day before the transport. At the start of the experiment (day 0), skin, gill and heart tissue samples (N = 6) were placed in sterile tubes containing 1 ml RNAlater. Total RNA from heart was then used for SAV analysis using RT-qPCR, while for the analysis of *Branchiomonas* and Salmon Gill Pox Virus (SGPV), gill samples were sent to Pharmaq analytiq, Bergen. For microbiome analyses, diploid and triploid salmon (N = 6) were sampled on days 0 and 21 post-infection. One $cm^2$ piece of the skin and one of the second gill arch were placed in tubes containing 1 ml sterile sucrose lysis buffer (SLB) for DNA extraction (Mitchell & Takacs-Vesbach, 2008). To confirm SAV infection, heart and pancreas tissues were sampled for histology 14 days post-infection, when PD pathology is most commonly observed [24]. Additionally, gill tissues (N = 3/group) were collected for histology on day 0 and 21 post-infection from the same individuals used for microbiome sequencing.

## DNA extraction, 16S rDNA PCR amplification, and sequencing

Whole genomic DNA was extracted from skin and gill samples by first lysing the tissue using sterile 3mm tungsten beads (Qiagen) in a Qiagen TissueLyser II, then followed by the cetyltri-methylammonium bromide method as previously described [25]. DNA was suspended in 30 μL RNase and DNase free molecular biology grade water and purity was assessed using a NanoDrop ND 1000 (Thermo Scientific).

   PCR was performed in triplicate on each sample, using primers targeting the V1-V3 region of the 16S rDNA marker gene. The primer sequences were as follows: 28F 5′-GAGTTTGAT CNTGGCTCAG-3′ and 519R 5′GTNTTACNGCGGCKGCTG-3′ (where N = any DNA nucle-otide, and K = T or G)[21]. 16S amplicons were generated by using Quantabio 5PRIME Hot-MasterMix and the following thermocycler conditions: 94° C for 90s; 33 cycles of 94° C for 30s, 52° C for 30s, 72° C for 90s; and a final extension of 72° C for 7 min. Amplicons were puri-fied using the Axygen AxyPrep Mag PCR Clean-up Kit (Thermo Scientific), and eluted into 30 μL molecular biology grade water. Unique oligonucleotide barcodes were ligated to the 5' and 3' ends of each sample, as well as the Nextera adaptor sequences, using the Nextera XT Index Kit v2 set A (Illumina). DNA concentrations were quantified using a Qubit, and normal-ized to a concentration of 200 ng/μL for DNA library pooling. Pooled samples were cleaned once more using the Axygen PCR clean-up kit before submitting them for sequencing. With each sequencing run, we included a mock community positive control consisting of equal amounts of DNA isolated from 7 bacterial cultures. Paired end sequencing was performed on the Illumina MiSeq platform using the MiSeq Reagent Kit v3 (600 cycle) at the Clinical and Translational Sciences Center at the University of New Mexico Health Sciences Center, gener-ating forward and reverse reads of 300 base pairs.

## Viral copy number quantification

SAV3 RNA from heart tissue was quantified with a modified one-step nsP-1 assay (Ag-Path, Ambion) [26] with a sense probe, using 200 ng total RNA in a total reaction volume of 10 μl. The detailed procedure for this analysis has been published earlier [22]. Briefly, heart tissue samples (half the heart) were homogenized in 1 ml TRIzol® and 450 μl of supernatants were used further for RNA isolation using a Purelink total RNA extraction kit, in an iPrep machine (Life Technologies). RNA was eluted in 50 μl of the propriety buffer, and concentration was estimated using a Nanodrop 1000 ND. Ten percent of the samples were checked for integrity on a Bioanalyser (Agilent Instruments) resulting in RINs ≥ 8. Approximately twenty percent of heart RNA samples randomly selected from all groups and time-points were qualitatively verified by measuring the transcription of elongation factor 1A [27]. cDNA was transcribed

from 200 ng total RNA in a 10 μl reaction using SuperScript™ VILO™ (Invitrogen) as described in the manufacturer's instructions. qPCR was run in triplicate in 96 well plates using Taq-Man® Fast Universal Master Mix (Applied Biosystems®) and an Applied Biosystems 7900H Fast sequence detection analyser. A 10 μl reaction volume contained; 2 μl cDNA (diluted 1:10), 5 μl 2 x master mix, 900nM of each primer and 250nM of FAM-labelled probe. The running conditions were as recommended by the manufacturer.

SAV3 RNA was quantified using a standard curve produced using a 10 x dilution series of synthetic SAV3 RNA (576 bps, cRNA) containing from 10 to 10 million copies of SAV RNA [15]. These standards were analyzed on each plate when quantifying copy numbers from heart RNA in the one-step assay.

C.Branchiomonas and SGPV quantification was carried out by RT-qPCR analysis at Pharmaq analytiq (Norway) using approved standardized tests.

### Histology

Gill tissues (N = 3) were fixed in 10% neutral buffered formalin, embedded in paraffin. Of the six individuals sampled for microbiome sequencing, three random individuals were chosen within each group for histopathological analysis. The embedded tissues were sectioned to 3 μm-thick sections and stained with Haematoxylin-Erythrosin-Saffron (HES) before visualizing under a Leica DMRBE light microscope (Leica Microsystems, Germany). The stained sections were scanned with a Hamamatzu NanoZomer S60, and photographs were taken using Spotflex camera model nr 15.2 64 Mp Shifting pixel (Diagnostic instruments Inc, USA) and processed with NDP.view2.

### Data analysis and statistics

Sequence data was analyzed using the latest version of Quantitative Insights into Microbial Ecology 2 (Qiime2 v2019.4) [28]. Demultiplexed sequence reads were preprocessed using DADA2, a plugin that supports quality filtering, denoising, merging paired ends, and removal of chimeric reads [29]. The first 35 base pairs were trimmed from forward and reverse reads before merging to remove adaptors. Amplicon sequence variants (ASVs) generated by DADA2 were assigned taxonomy by aligning to the latest version of the Silva 16S rDNA database (v132). Samples were rarefied before core diversity analyses, which included total number of observed ASV's, Shannon's Diversity Index, Chao1. A mixed model ANOVA was used to test the effects of all possible interactions (ploidy*treatment*time) on the alpha diversity parameters Chao1, Shannon Diversity and Observed ASVs using RStudio version 1.3.959. Normal distribution of residuals was first confirmed by Shapiro-Wilk normality test in R prior to running the mixed model ANOVA analysis.

## Results

### High throughput sequencing analysis

A total of 6,591,584 raw reads were obtained from all skin samples. After quality filtering with DADA2 and removal of non-specific salmon genomic reads, there remained 1,944,093 reads, with a mean of 54,002 reads per sample. For core diversity analysis, samples were rarefied to a sampling depth of 14,270.

A total of 7,044,068 raw reads were obtained from all gill samples. Quality filtering with DADA2 and removal of salmon genomic reads left 554,903 reads, with a mean of 15,414 per sample. For core diversity analysis, samples were rarefied to 1830 reads per sample, which excluded one diploid day 0 control sample and two triploid day 0 control samples.

**Table 1. Mixed model ANOVA analysis of Shannon Diversity Index, and Chao1 in skin of diploid and triploid salmon.**

| Source | Diversity index | SE | t value | *P*-value |
|---|---|---|---|---|
| Ploidy | Shannon | 0.1424 | 0.343 | 0.7336 |
| | Chao1 | 0.1134 | 0.366 | 0.7169 |
| Time | Shannon | 0.0068 | -0.189 | 0.8509 |
| | Chao1 | 0.0063 | -3.876 | 0.0005 *** |
| Treatment | Shannon | 0.1424 | -3.091 | 0.00428 ** |
| | Chao1 | 0.1819 | -3.856 | 0.0005 *** |
| Ploidy:Time | Shannon | 0.0096 | -2.901 | 0.0069 ** |
| | Chao1 | 0.0100 | -3.149 | 0.0036 ** |
| Ploidy:Treatment | Shannon | 0.2014 | 4.716 | $5.18e^{-5}$ *** |
| | Chao1 | 0.2501 | 6.042 | $1.24e^{-6}$ *** |
| Time:Treatment | Shannon | NA | NA | NA |
| | Chao1 | NA | NA | NA |
| Ploidy:Time:Treatment | Shannon | NA | NA | NA |
| | Chao1 | NA | NA | NA |

## Alpha diversity of diploid and triploid salmon microbiomes

Mixed model ANOVA results indicate the "ploidy" had no significant effect on the alpha diversity (Shannon diversity index and Chao 1) of the Atlantic salmon skin microbial communities of Atlantic salmon. "Time" significantly affected Chao 1 of the skin microbial community whereas "treatment" significantly impacted both Shannon diversity index and Chao1 (Table 1). The interactions between ploidy and time as well as ploidy and treatment significantly impacted Shannon Diversity and Chao1 in the skin (Table 1).

In the gills, "ploidy" and "time" were both significant determining factors of the Shannon Diversity Index but not for Chao1, while "treatment" did not significantly affect any alpha diversity metrics (Table 2). We also observed a significant "Ploidy:time" interaction for Shannon Diversity index but not Chao1 in the gills (Table 2). Combined, these data indicate that ploidy had a greater impact on the gill compared to the skin microbial community and that SAV infection did not alter alpha diversity metrics in the present study. Further, ploidy

**Table 2. Mixed model ANOVA analysis of Shannon Diversity Index, and Chao1 in gills of diploid and triploid salmon.**

| Source | Diversity index | SE | t value | *P*-value |
|---|---|---|---|---|
| Ploidy | Shannon | 0.2496 | 4.725 | $6.37e^{-5}$ *** |
| | Chao1 | 0.2886 | 1.188 | 0.244 |
| Time | Shannon | 0.0107 | -2.439 | 0.0216 * |
| | Chao1 | 0.0176 | -1.631 | 0.113 |
| Treatment | Shannon | 0.2148 | -0.802 | 0.4296 |
| | Chao1 | 0.4726 | -0.858 | 0.398 |
| Ploidy:Time | Shannon | 0.0156 | -6.234 | $1.14e^{-6}$ *** |
| | Chao1 | 0.0240 | -0.520 | 0.607 |
| Ploidy:Treatment | Shannon | 0.3038 | 0.488 | 0.6294 |
| | Chao1 | 0.6470 | 0.142 | 0.888 |
| Time:Treatment | Shannon | NA | NA | NA |
| | Chao1 | NA | NA | NA |
| Ploidy:Time:Treatment | Shannon | NA | NA | NA |
| | Chao1 | NA | NA | NA |

interaction with time was a significant determinant of the alpha diversity of both the skin and gill microbial community.

## Skin microbial community composition

The skin microbial community composition was nearly identical between diploid and triploid salmon (PERMANOVA adjusted $P$-value = 0.589). The three most dominant phyla represented in the skin were Firmicutes (44.79% in diploids, 42.75% in triploids), Bacteriodetes (29.64% in diploids, 29.29% in triploids), and Proteobacteria (22.33% in diploids, 23.69% in triploids), with low levels of Planctomycetes (2.24% in diploids, 3.28% in triploids) present in all samples (Fig 1A). At the genus level, most Firmicutes reads were identified as *Paenibacillus* sp., which contributed to 44.44% and 42.5% of the overall diversity in diploids and triploids, respectively. *Hydrotalea* sp. was the second most abundant genera accounting for 22.96% and 22.56% of the overall diversity in diploids and triploids, respectively, followed by an unidentified member of the family Burkholderiaceae (13.45% in diploids, 13.82% in triploids) (Fig 1B). Differential abundance testing with ANCOM showed that no ASVs were differentially abundant between diploid and triploid skin samples.

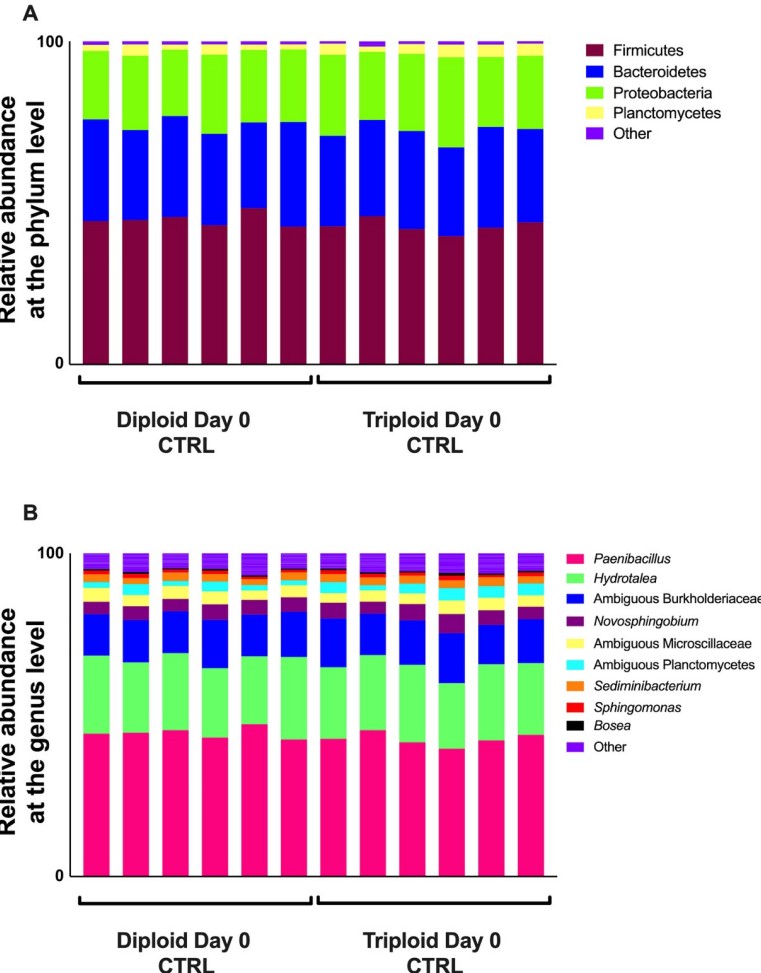

**Fig 1. Microbial composition in the skin of triploid and diploid Atlantic salmon at the steady state.** (A) Relative abundance at the phylum level for triploid and diploid skin at day 0. (B) Relative abundance at the genus level for triploid and diploid skin at day 0.

## Gill microbial community composition

We observed some differences in the microbial communities associated with the gills of diploid and triploid salmon, though there were no significant differences in beta diversity between the gills of triploid and diploid salmon (PERMANOVA adjusted $P$-value = 0.133). The gill microbial community of Atlantic salmon was composed almost entirely of Proteobacteria. Proteobacteria accounted for 99.19% of all diversity present in the gills of diploid salmon and 91.94% in triploid salmon (Fig 2A). There was a larger representation of Firmicutes in 4 of 6 triploids, resulting in this phylum on average accounting for 7.51% of all triploid uninfected controls, and just 0.5% of all diploid uninfected controls (Fig 2A). At the genus level (Fig 2B), many samples were marked by an abundance of *C. Branchiomonas*. This taxon was more prevalent in diploids, representing 77.3% of microbial diversity, compared to 24.2% in triploids. The next most represented genera were *Oleispira* (9.58% in diploids, 49.04% in triploids), followed by *Moritella* (0.2% in diploids, 7.81% in triploids), *Staphylococcus* (0.57% in diploids, 5.57% in triploids), and *Acidovorax* (0.88% in diploids, 2.57% in triploids) (Fig 2B). Differential abundance testing with ANCOM showed *C. Branchiomonas* as the only differentially abundant ASV.

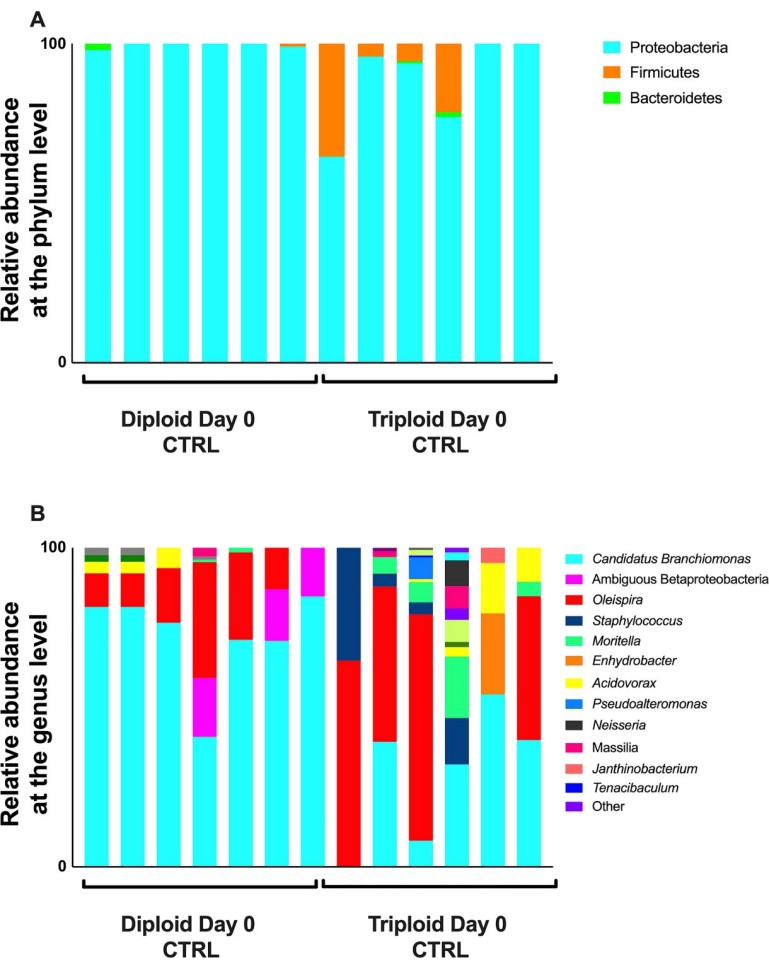

**Fig 2. Relative microbial composition in the gills of triploid and diploid Atlantic salmon at the steady state.** (A) Relative abundance at the phylum level for triploid and diploid gills at day 0. (B) Relative abundance at the genus level for triploid and diploid gills at day 0.

## Effects of SAV3 bath challenge on skin and gill microbiomes

We next sought to investigate how bath challenge with SAV3 would impact the skin and gill microbiome of diploid and triploid Atlantic salmon. We collected skin and gill samples 21 days post SAV challenge from six bath-challenged fish from each group and compared to unchallenged controls from the same time-point. SAV3 infection in challenged fish was validated by RT-qPCR on heart samples using qPCR of the nsP1 gene (Table 3).

Significant differences in alpha diversity were observed between control and infected groups for both diploids and triploids (Fig 3A–3C). Specifically, the total number of observed ASVs, Shannon Diversity Index and Chao1 values were highest in the diploid day 0 control group and significantly decreased in diploid control group by day 21 (Fig 3A–3C). Diploid infected fish at day 21 had significantly lower alpha diversity values than diploid control day 21 salmon. The same was true for triploid fish which showed the highest skin alpha diversity metrics in control animals at day 0 with values significantly dropping in triploid control animals at day 21 and reaching the lowest value in the SAV3-infected day 21 group.

In the gills a similar trend to that found in the skin was observed with some differences. The total number of ASVs was highest in control day 0 diploid and triploid animals and significantly decreased in control day animals (Fig 3D). However, no changes were observed between day 21 control and day 21 infected groups, neither in diploids nor in triploids (Fig 3D). Shannon Diversity Index was only significantly different between the control triploid day 0 and the control triploid day 21 groups, with a significant drop over time (Fig 3E). No significant differences were found in gill Chao 1 values throughout the experiment yet the trend was the same as that observed for the other two alpha diversity metrics in gill samples (Fig 3F).

**Table 3. SAV3 levels in heart tissue samples from fish used in this study.** SAV3 titers were estimated using qPCR on nsP1 gene. ND—not detected (below detection limit).

| Fish ID | Condition, Time, Treatment | nsP1 copy number |
|---------|---------------------------|------------------|
| 3–18 | Diploid Day 21 CTRL | ND |
| 3–19 | Diploid Day 21 CTRL | ND |
| 3–20 | Diploid Day 21 CTRL | ND |
| 4–17 | Diploid Day 21 CTRL | ND |
| 4–18 | Diploid Day 21 CTRL | ND |
| 4–19 | Diploid Day 21 CTRL | ND |
| 7–18 | Diploid Day 21 SAV | 424394.22 |
| 7–19 | Diploid Day 21 SAV | 21167.836 |
| 7–20 | Diploid Day 21 SAV | 6001459.5 |
| 8–17 | Diploid Day 21 SAV | 505549.8 |
| 8–18 | Diploid Day 21 SAV | 84178.266 |
| 8–19 | Diploid Day 21 SAV | 44352.566 |
| 5–18 | Triploid Day 21 CTRL | ND |
| 5–19 | Triploid Day 21 CTRL | ND |
| 5–20 | Triploid Day 21 CTRL | ND |
| 6–17 | Triploid Day 21 CTRL | ND |
| 6–18 | Triploid Day 21 CTRL | ND |
| 6–19 | Triploid Day 21 CTRL | ND |
| 1–18 | Triploid Day 21 SAV | 25218.408 |
| 1–19 | Triploid Day 21 SAV | 27687.645 |
| 1–20 | Triploid Day 21 SAV | 2188644.2 |
| 2–17 | Triploid Day 21 SAV | ND |
| 2–18 | Triploid Day 21 SAV | ND |
| 2–19 | Triploid Day 21 SAV | ND |

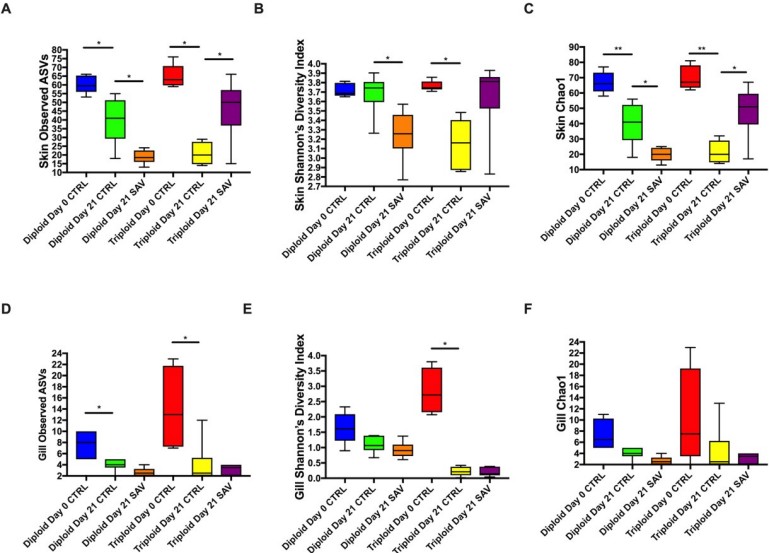

**Fig 3. Alpha diversity metrics for the skin and gill microbiome of triploid and diploid control and infected Atlantic salmon.** (A) Total number of observed ASVs in the skin. (B) Shannon's diversity index in the skin. (C) Chao1 index in the skin. (D) Total number of observed ASVs in the gills. (E) Shannon's diversity index in the gills. (F) Chao1 index in the gills.

Mixed model ANOVA analyses indicated that significant effects due to SAV infection ("treatment") on skin but not gill alpha diversity values (Tables 1 and 2). There was also a significant interaction between "ploidy" and "treatment" for both Shannon Diversity Index and Chao1 values of the skin but not the gill microbial community. Further, we observed significant differences in skin beta diversity in diploid salmon in response to SAV infection (PERMANOVA adjusted $P$-value = 0.028), though these differences were not significant in triploids (PERMANOVA adjusted $P$-value = 0.14). Differences observed between the control and infected diploid treatments were due to low abundant taxa and therefore, overall, the community composition of control and infected groups were largely similar (Fig 4).

In the gills at day 21, *C. Branchiomonas* was the dominant taxon in both diploid and triploid salmon, whether they were control or infected. This taxon made up 99.13% of the gill microbiome in uninfected diploid fish and 65.39% in infected diploids by 21 days post infection. This

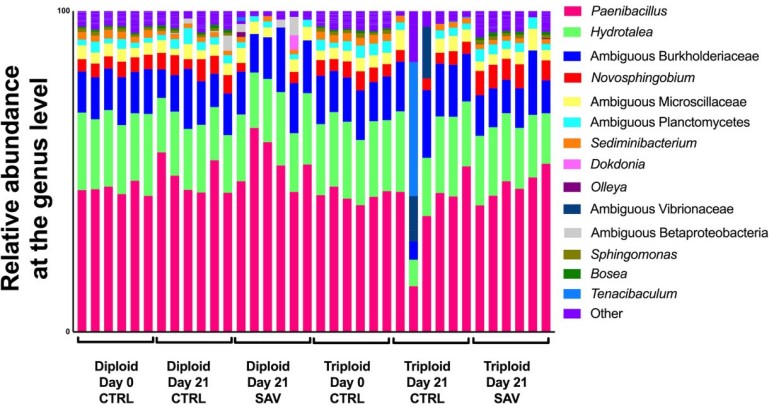

**Fig 4. Relative microbial composition in the skin of Atlantic salmon following challenge with SAV3.** Relative abundance at the genus level for triploid and diploid skin at day 0, day 21 unchallenged, and day 21 post-immersion in SAV bath challenge.

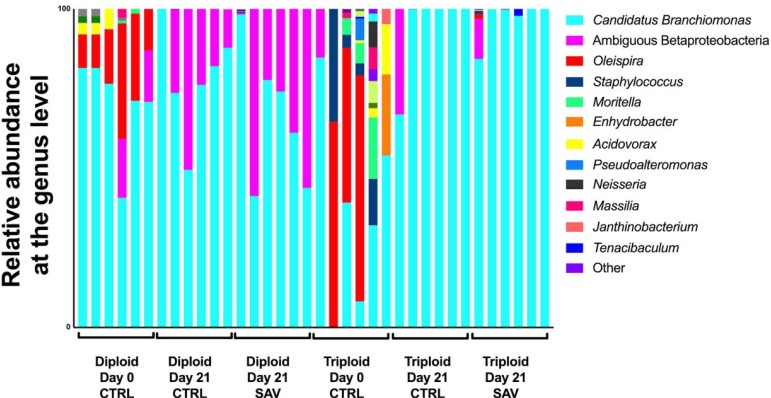

**Fig 5. Relative microbial composition in the gills of Atlantic salmon following challenge with SAV3.** Relative abundance at the genus level for triploid and diploid gills at day 0, day 21 unchallenged, and day 21 post-immersion in SAV bath challenge.

pattern was reversed in triploid salmon, as *C. Branchiomonas* accounted for 77.35% of the gill microbiome in uninfected fish and 99.62% in infected salmon at day 21 post infection (Fig 5).

The expansion of *C. Branchiomonas* therefore resulted in significantly decreased alpha diversity measures for all groups at day 21(Fig 5), compared to day 0 controls, though no significant differences in ASV were observed between day 21 control and infected groups for either diploids or triploids (Fig 3D–3F). Additionally, there were no significant differences in beta diversity for both diploids and triploids in response to SAV3 infection (PERMANOVA adjusted *P*-value = 0.364 for diploids, PERMANOVA adjusted *P*-value = 0.777 for triploids).

In order to confirm the 16S rDNA results, the relative abundance of *C. Branchiomonas* was analyzed using qPCR. Analysis of *C. Branchiomonas* in gill tissue samples from fish before transport to the experimental facility showed significantly lower abundance in triploids (Average Ct value±SE = 35,2±6.8) compared to diploids (Average Ct value ±SE = 24.7±3.2) (*P*-value = 0.002) (Fig 6). At Day 0, the abundance of *C. Branchiomonas* increased in both diploids

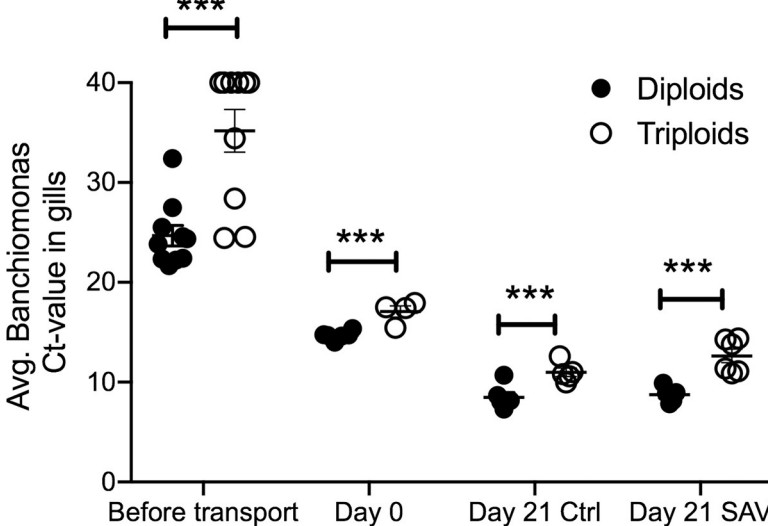

**Fig 6. Mean *Candidatus Branchiomonas* Ct values in gill samples of diploid and triploid Atlantic salmon as determined by RT-qPCR.** *** indicates *P*-values<0.001 by Paired Student's T-test.

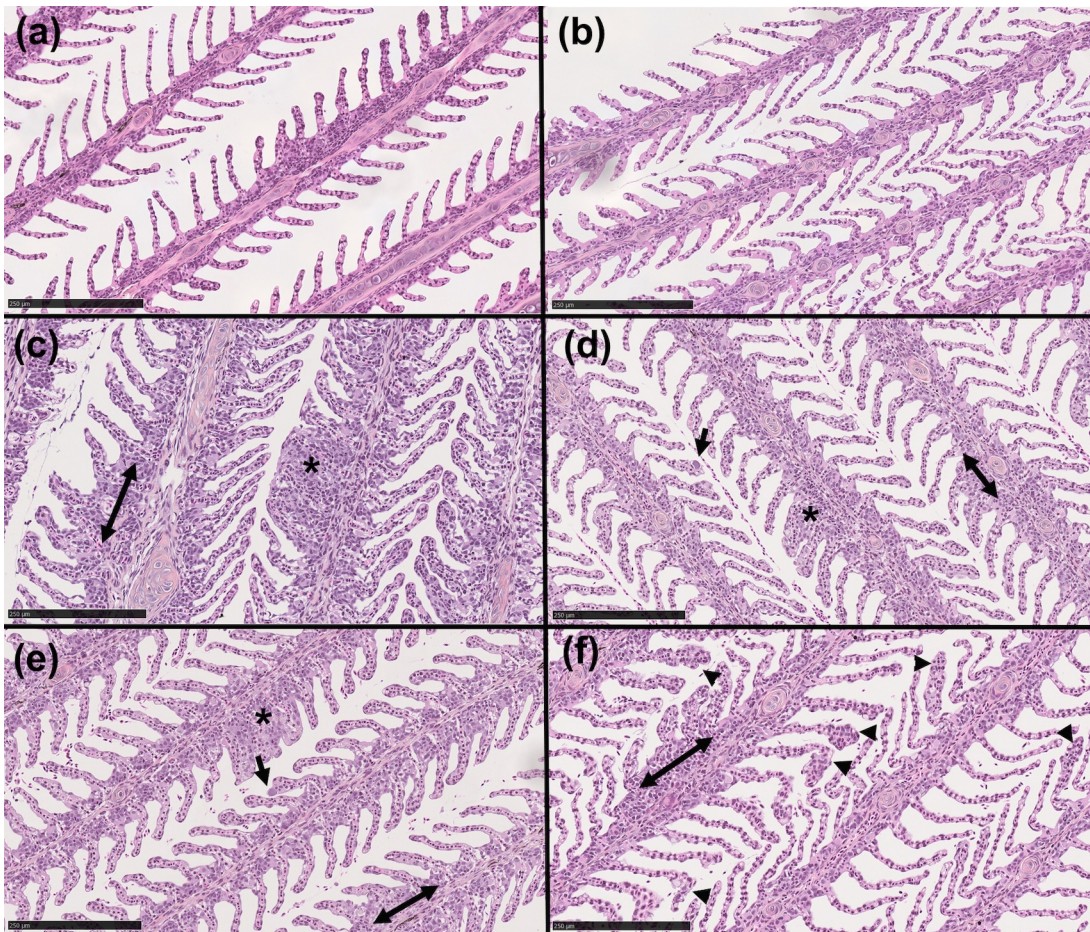

**Fig 7. Histological examination of the gills of triploid and diploid Atlantic salmon.** (A, B) Hematoxylin & eosin (H&E) stain of diploid and triploid day 0 control gill. (C, D) H&E stain of diploid and triploid day 21 control gill. (E, F) H&E stain of diploid and triploid day 21 SAV infected salmon gill. Arrows indicate epitheliocystis; arrowheads indicate thickening of secondary lamellae tips; double arrows indicate enlargement of basal region of lamellae; asterisks indicate fusion of secondary lamellae. Images are representative of N = 3/group. Scale bar: 250 μm.

and triploids but remained significant lower in triploids (Average Ct value±SE = 17.1±1.1) compared to diploids (Average Ct value±SE = 14.7±0.44) (*P*-value = 0.018). At day 21, *C. Branchiomonas* levels continued to increase in both control and SAV infected groups but levels were still significantly lower in triploid than in diploid fish (Fig 6).

Histological analyses of the gills showed healthy gill morphology for both diploid and triploid day 0 samples (Fig 7A and 7B). All day 21 samples were visibly less healthy than day 0 controls, regardless of SAV infection. All day 21 groups frequently showed fusion of the secondary lamellae, thickening of basal region of lamellae, and necrosis (Fig 7C–7F). We also observed signs of gill epitheliocystis sporadically throughout the day 21 samples, a symptom of *C. Branchiomonas* infection. In diploid fish, epitheliocystis was only observed in day 21 infected samples, and not in uninfected controls (Fig 7C and 7E). In triploid samples, some signs of epitheliocystis were observed in both day 21 control and infected fish, though it was more severe in the controls (Fig 7D and 7F).

## Discussion

Using sterile, triploid Atlantic salmon in commercial aquaculture reduces the threat of farmed, escaped salmon breeding with wild salmon populations. Triploid salmon, although phenotypically indistinguishable from diploids, have shown physiological differences requiring optimization of rearing conditions compared to diploids. This has manifested itself as a relative intolerance to warmer seawater temperatures compared to diploid salmon [30], and as such the bulk production of triploid salmon takes place in the cooler northern Norwegian waters. Another challenge when rearing triploid salmon has been optimizing food micronutrients to overcome skeletal deformities and cataracts that are more common in triploid salmon [4, 5].

The use of triploid salmon in aquaculture is being explored, however, a full characterization of triploid fish physiology including disease resistance traits, responses to stress and immune responses is necessary in order to predict the outcomes of using triploid fish in different settings. The microbiome influences every aspect of host physiology including metabolism, development, reproduction, immunity, movement and behavior [31–34]. Previous studies have determined that microbiome assemblies are influenced by a number of factors including environment, diet and host genetics [35, 36]. Yet, the microbial communities living in association with the external surfaces of diploid and triploid salmon and how they respond to infection remain unknown.

Our results indicate that at the steady state, the skin microbial community of triploid and diploid salmon is largely the same and therefore, the environment largely shapes these communities with little contribution from host genetics factors. In support, other fish microbiome studies have reached similar conclusions [37, 38] highlighting a predominant role of environment over host genetics in fish microbial community assembly. The top three represented phyla, Firmicutes, Proteobacteria, and Bacteriodetes, were also among the top represented phyla observed in the skin of Atlantic salmon post-smolts from other studies [19, 21]. In a previous study we reported shifts in the skin microbiome of diploid salmon in response to low or high doses of SAV3 in a similar bath challenge [21]. The present study used an intermediate dose of the virus but appears inconsistent by comparison. However, these experiments used different, fish, experimental conditions and sampling time-points and therefore comparisons are not straightforward.

In the gills, *Candidatus Branchiomonas* was the dominant taxon across all groups. *C. Branchiomonas* is the causative agent of gill epitheliocystis in Atlantic salmon [39] but the extent to which *C. Branchiomonas* contributes to gill disease is still unclear. While studies have suggested that epitheliocystis caused by *C. Branchiomonas* is correlated with proliferative gill inflammation [40] this bacterium has been detected as a member of the normal gill microbiome of healthy salmonids [41, 42]. At the steady state, we found that *C. Branchiomonas* was more abundant in diploid compared to triploid salmon although inter-individual variation was observed. Interestingly, *Oleispira* sp., a bacterium previously identified in the skin of Atlantic salmon and thought to be important in the salmon smoltification process [19, 21], was more abundant in triploid than diploid salmon suggesting that displacement of *Oleispira* sp. by *C. Branchiomonas* may occur.

Despite the detection of *C. Branchiomonas* by 16S rDNA sequencing in day 0 controls, histology showed no signs of disease in the gills of these fish. This finding supports the idea that *C. Branchiomonas* can be part of the normal gill microbiome of salmonids without causing disease and/or that presence of the bacterium precedes appearance of histopathology. Three weeks later, the abundance of this pathogen increased in all groups indicating the presence of a natural infection in our study. Hallmarks of *C. Branchiomonas* infection were however observed differentially in triploid and diploid fish. Instances of epitheliocystis were frequently

observed in the gills of SAV infected diploids but were not found in the day 21 control group. Meanwhile, in triploids, although abundance of *C. Branchiomonas* RNA was lower than in diploids, epitheliocystis was observed in gills from both day 21 control and infected fish.

Low levels of *C. Branchiomonas* were observed in the gills of triploid and diploid salmon sampled before transport to our experimental facilities. Fish are frequently transported in aquaculture settings, and it is established that the effects of transportation stress disrupt skin homeostasis [19], and thus it is plausible that the expansion of *C. Branchiomonas* by the start of the experiment was influenced by effects of transportation stress. Though we did not initially anticipate such high loads of *C. Branchiomonas*, in our study, these results provide an opportunity to examine how triploid fish respond to co-infection. Co-infections are common in farmed salmonid species [43–45], and these concomitant infections can have varied effects on host immune responses. Putative pathogens that are associated with gill disease were detected in parallel in a screening of Atlantic salmon sampled from an offshore farm in Ireland from the production cycle in 2012–2014 [46]. These pathogens included Salmon gill poxvirus (SGPV), *Neoparamoeba peruans*, *C. Branchiomonas*, *Tenacibaculum maritimum*, and the microsporidian *Dezmozoon lepeophtherii*. Though SGPV was not detected in any of our samples, it appears that co-infection with SAV may contribute to gill disease. It is also possible that additional pathogens apart from the three we screened for in this study (SAV, SGPV, *C. Branchiomonas*) were present in our samples. Downes et al., [46] found accounts of epitheliocystis inconsistently throughout samples, though modeling of gill histopathology scores showed *C. Branchiomonas* (as well as *N. peruans*) to have a meaningful association with gill histopathology score, and suggested that *C. Branchiomonas* may even be protective in this regard [46]. Our results demonstrated that symptoms of gill disease were observed in samples with high loads of *C. Branchiomonas*, even though SAV exposed fish were not SAV positive. In diploids, co-infection with these two pathogens resulted in the presence of epitheliocystis and more severe histopathology scores. In triploids, on the other hand, there were no clear differences in occurrence of epitheliocystis or histopathology score upon co-infection.

It is well established that triploid Atlantic salmon display problems associated with skeletal deformations, growth, and survival- though these problems can be corrected through refined husbandry [4, 5]. Although the fish in this study received standard feed for diploid fish, which is sub-optimal for triploid fish, and the experiment was carried out at sub-optimal temperature there was no detrimental effect on the maximal prevalence of SAV3 in the triploid group compared to the diploid group. In addition to skeletal deformities, triploid Atlantic salmon can also suffer from gill filament deformity syndrome (GFD), which is characterized by missing primary gill filaments [4, 5]. Interestingly, triploid salmon have a reduced gill surface area compared to diploids, which may contribute to deficiencies in nutrient uptake and osmoregulation, resulting in impaired gill health. Despite these observations, a study [47] showed no significant effects of ploidy on mortality following experimental infection with *N. perurans*. This study did show reduced lysozyme activity in triploid fish, which may be relevant for dealing with intracellular pathogens such as *C. Branchiomonas*. Ploidy associated differences in response to SAV infection in Atlantic salmon have been documented, as triploids accumulated prevalence more slowly than diploids after a bath challenge [22]. Our results showed slightly increased gill histopathology scores in triploids compared to diploids, as well as increased accounts of epitheliocystis, despite a lower abundance of *C. Branchiomonas* detected by qPCR.

In conclusion, the present study demonstrates little effects of ploidy on the skin and gill microbial communities of Atlantic salmon and very little effects of SAV experimental infection on these communities. Interestingly, our sequencing efforts detected the high prevalence of a relevant gill pathogen in salmon aquaculture, *C. Branchiomonas* in the gill microbial

communities of both diploid and triploid animals. Our findings therefore provide new insights into gill health of diploid and triploid salmon in the presence of multiple pathogenic stressors.

## Acknowledgments

Authors wish to thank Dr. Darrel Dinwiddie for sharing the Illumina sequencing instrument.

## Author Contributions

**Conceptualization:** Sonal Patel, Irene Salinas.

**Formal analysis:** Ryan Brown, Amir Mani, Sonal Patel.

**Funding acquisition:** Sonal Patel, Irene Salinas.

**Investigation:** Ryan Brown, Lindsey Moore, Sonal Patel, Irene Salinas.

**Methodology:** Ryan Brown, Lindsey Moore.

**Supervision:** Irene Salinas.

**Validation:** Amir Mani.

**Visualization:** Ryan Brown.

**Writing – original draft:** Ryan Brown, Irene Salinas.

**Writing – review & editing:** Ryan Brown, Lindsey Moore, Amir Mani, Sonal Patel, Irene Salinas.

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
