## [Decision Letter · Decision Letter 0]

15 Jan 2021

PONE-D-20-37068

Effects of ploidy and salmonid alphavirus infection on the skin and gill microbiome of Atlantic salmon (Salmo salar)

PLOS ONE

Dear Dr. Salinas,

Thank you for submitting your manuscript to PLOS ONE. After careful consideration, we feel that it has merit but does not fully meet PLOS ONE’s publication criteria as it currently stands. Therefore, we invite you to submit a revised version of the manuscript that addresses the points raised during the review process.

You will see that reviewers did not raise  major issues; please address all questions, and clarify  points mentioned by reviewer #1.

We look forward to receiving your revised manuscript.

Kind regards,

Pierre Boudinot, phD

Academic Editor

PLOS ONE

Journal Requirements:

2. We note that you are reporting an analysis of a microarray, next-generation sequencing, or deep sequencing data set. PLOS requires that authors comply with field-specific standards for preparation, recording, and deposition of data in repositories appropriate to their field. Please upload these data to a stable, public repository (such as ArrayExpress, Gene Expression Omnibus (GEO), DNA Data Bank of Japan (DDBJ), NCBI GenBank, NCBI Sequence Read Archive, or EMBL Nucleotide Sequence Database (ENA)). In your revised cover letter, please provide the relevant accession numbers that may be used to access these data. For a full list of recommended repositories, see http://journals.plos.org/plosone/s/data-availability#loc-omics or http://journals.plos.org/plosone/s/data-availability#loc-sequencing.

Reviewers' comments:

Reviewer's Responses to Questions

**Comments to the Author**

1. Is the manuscript technically sound, and do the data support the conclusions?

Reviewer #1: Yes

Reviewer #2: Yes

2. Has the statistical analysis been performed appropriately and rigorously? 

Reviewer #1: Yes

Reviewer #2: Yes

3. Have the authors made all data underlying the findings in their manuscript fully available?

Reviewer #1: Yes

Reviewer #2: Yes

4. Is the manuscript presented in an intelligible fashion and written in standard English?

Reviewer #1: Yes

Reviewer #2: Yes

5. Review Comments to the Author

Reviewer #1: The manuscript evaluates the effects of ploidy on skin and gill microbiome and pathology of Altantic salmon, challenged with SAV3 and C. Branchiomonas. The study presents very interesting results for salmon aquaculture, with regard to the interaction of ploidy and gill health as well the resistance to pathogenic outbreaks. The manuscript is well written, with the proper experimental design and conclusions supported by the results.

There are several clarifications that are still required, as presented below:

Methods:

Line 110-114: There is some missing information in the text. The authors describe the preparation of the fish to enter to the smolt stage, that involves living in sea water. Then they describe that fish were transfer one month later to freshwater. During this month that the authors describe, were the fish transferred in seawater and then freshwater? The further trials were performed in seawater, so I think the way is written is a bit confusing.

Line 131-132: How could the authors monitor the concentration of the infectious virus during the challenge? At which concentrations were the fish exposed to?

Line 138: What was the reason that the fish were exposed to a temperature of 14 and not 12 degrees, like in the beginning of the experiment?

Line 155: Please explain here what SGPV stands for. Did the authors expect a natural outbreak due to the SAV3 infection?

Results:

For all baplot figures: The y axis values are all missing.

Line 254: Why did the authors subsampled the data in such a low sequencing depth?

Line 353-359: A good way to represent overall changes in beta diversity between the treatments would be by Principal component analysis, supported also by Permanova. Did they authors perform such analysis?

Line 377-378: The authors could potentially also show such an effect by a correlation analysis, but this is only a suggestion.

Line 401-403: Which parameters were used to assess those observations? Did the authors use any statistical tests to evaluate the histological observations?

Reviewer #2: The manuscript titled “Effects of ploidy and salmonid alphavirus infection on the skin and gill microbiome of Atlantic salmon (Salmo salar)” reports the effects of ploidy on the skin and gill microbial communities of Atlantic salmon, as well as the effect of SAV on these microbiomes. The investigation is really well planned and described, and the topic is really worthy for investigation, since the importance of the triploid Atlantic salmon culture.

The data support the conclusions, and the discussion is perfectly developed.

The manuscript is perfectly suitable for publication.

I have only some minor comments:

- The authors should check along the text “sp.” that it should not be in italic.

- Line 124: For the experimental SAV3 infection, the viral titer should be included.

- Line 155: Include Salmon Gill Pox Virus, since it is the first time that is mentioned.

- Lines 349-350: It is not clear that there were no significant effects on any alpha diversity value due to the treatment (SAV infection), since in skin (Table 1) significant differences are shown.

- The results represented in Fig3 should be described with more detail.

6. PLOS authors have the option to publish the peer review history of their article (what does this mean?). If published, this will include your full peer review and any attached files.

Reviewer #1: No

Reviewer #2: No

---

## [Author Response · Author response to Decision Letter 0]

21 Jan 2021

Albuquerque January 19th 2021

PONE-D-20-37068

Effects of ploidy and salmonid alphavirus infection on the skin and gill microbiome of Atlantic salmon (Salmo salar)

Thank you to the reviewers for their time and feedback provided. Please find below a point-by-point answer to each of the comments raised by both reviewers.

Journal Requirements:

2. We note that you are reporting an analysis of a microarray, next-generation sequencing, or deep sequencing data set. PLOS requires that authors comply with field-specific standards for preparation, recording, and deposition of data in repositories appropriate to their field. Please upload these data to a stable, public repository (such as ArrayExpress, Gene Expression Omnibus (GEO), DNA Data Bank of Japan (DDBJ), NCBI GenBank, NCBI Sequence Read Archive, or EMBL Nucleotide Sequence Database (ENA)). In your revised cover letter, please provide the relevant accession numbers that may be used to access these data. For a full list of recommended repositories, see http://journals.plos.org/plosone/s/data-availability#loc-omics or http://journals.plos.org/plosone/s/data-availability#loc-sequencing.

The cover letter includes the accession number for our BioProject.

Reviewer #1: The manuscript evaluates the effects of ploidy on skin and gill microbiome and pathology of Altantic salmon, challenged with SAV3 and C. Branchiomonas. The study presents very interesting results for salmon aquaculture, with regard to the interaction of ploidy and gill health as well the resistance to pathogenic outbreaks. The manuscript is well written, with the proper experimental design and conclusions supported by the results.

There are several clarifications that are still required, as presented below:

Methods:

Line 110-114: There is some missing information in the text. The authors describe the preparation of the fish to enter to the smolt stage, that involves living in sea water. Then they describe that fish were transfer one month later to freshwater. During this month that the authors describe, were the fish transferred in seawater and then freshwater? The further trials were performed in seawater, so I think the way is written is a bit confusing.

Thank you. We have rephrased, lines 110 - 118. Parr-smolt transformation usually takes 6 weeks when fish are maintained in freshwater, but light regime artificially induces the changes to prepare salmon for their life in seawater.

Line 131-132: How could the authors monitor the concentration of the infectious virus during the challenge? At which concentrations were the fish exposed to?

Yes, the amount of virus was checked in the water the fish were to be exposed in prior to the start of the experiment. More information has been added in Line 133, and lines 143-148.

Line 138: What was the reason that the fish were exposed to a temperature of 14 and not 12 degrees, like in the beginning of the experiment?

Thank you for bringing up this point. We chose 14C for the infection because outbreaks of SAV3 in Norway mostly occur during the period Spring – late Autumn and peak when water temperature in the sea is above 12 degrees (Fiskehelserapporten_2019_web.pdf; figure for outbreak of SAV3 in Norway). Since triploid salmon is being considered for production, it was relevant to carry out the experimental infection at 14 degrees to reflect summer/autumn field temperatures.

Line 155: Please explain here what SGPV stands for. Did the authors expect a natural outbreak due to the SAV3 infection?

Thank you, SGPV is now written in full. An SGPV outbreak was not expected. We checked SGPV levels because given the high presence of Branchiomonas sp. we wanted to rule out any presence of any other gill pathogens such as SGPV.

Results:

For all barplot figures: The y axis values are all missing.

Thank you. All bar plots are percentages, from 0 to 100. We have revised the figures.

Line 254: Why did the authors subsampled the data in such a low sequencing depth?

Sub-sampling at that depth was performed to be sure we could use as many of our samples as possible with high quality reads. We checked the rarefaction curves and all samples reached aysmptote at the choses sampling depth so we are sure they were not undersampled. 

Line 353-359: A good way to represent overall changes in beta diversity between the treatments would be by Principal component analysis, supported also by Permanova. Did they authors perform such analysis?

Yes, we did principal component analyses in all our datasets which was helpful to identify if there were any outliers. We did not include the graphs in the manuscript because this visualization did not really show any additonal useful information that was not already shown in the rest of the figures. 

Line 377-378: The authors could potentially also show such an effect by a correlation analysis, but this is only a suggestion.

Thank you, we think that the data as plotted already shows the biological meaning of our findings. 

Line 401-403: Which parameters were used to assess those observations? Did the authors use any statistical tests to evaluate the histological observations?

We performed a qualitative score for the histopathology but we did not carried out any statistical analysis or correlations on the histology data because only three animals were examined in each group therefore the analysis would be under-powered. We have added N=3 to te materials and methods and Figure legend since that information was missing.

Reviewer #2: The manuscript titled “Effects of ploidy and salmonid alphavirus infection on the skin and gill microbiome of Atlantic salmon (Salmo salar)” reports the effects of ploidy on the skin and gill microbial communities of Atlantic salmon, as well as the effect of SAV on these microbiomes. The investigation is really well planned and described, and the topic is really worthy for investigation, since the importance of the triploid Atlantic salmon culture.

The data support the conclusions, and the discussion is perfectly developed.

The manuscript is perfectly suitable for publication.

I have only some minor comments:

- The authors should check along the text “sp.” that it should not be in italic.

Thank you, this is now corrected.

- Line 124: For the experimental SAV3 infection, the viral titer should be included.

Thank you, we have included this information in lines Line 133, and lines 143-148.

- Line 155: Include Salmon Gill Pox Virus, since it is the first time that is mentioned.

It is now written in full, thank you.

- Lines 349-350: It is not clear that there were no significant effects on any alpha diversity value due to the treatment (SAV infection), since in skin (Table 1) significant differences are shown.

Thank you for picking up the discrepancy. There is an effect of treatment (SAV infection) on the skin alpha diversity but not in the gills. And this was already described in the results section of the original submission. We have fixed Lines 349-350 in the text, which now appears in lines 384-385.

- The results represented in Fig 3 should be described with more detail.

Thank you. We have expanded the text describing Figure 3 as suggested (see lines 358-375).

---

## [Editor Report · Decision Letter 1]

2 Feb 2021

Effects of ploidy and salmonid alphavirus infection on the skin and gill microbiome of Atlantic salmon (Salmo salar)

PONE-D-20-37068R1

Dear Dr. Salinas,

We’re pleased to inform you that your manuscript has been judged scientifically suitable for publication and will be formally accepted for publication once it meets all outstanding technical requirements.

Kind regards,

Pierre Boudinot, phD

Academic Editor

PLOS ONE

Additional Editor Comments

Please replace "the" by "then" lines 133 and 135.

---

## [Editor Report · Acceptance letter]

3 Feb 2021

PONE-D-20-37068R1 

Effects of ploidy and salmonid alphavirus infection on the skin and gill microbiome of Atlantic salmon (*Salmo salar*) 

Dear Dr. Salinas:

I'm pleased to inform you that your manuscript has been deemed suitable for publication in PLOS ONE. Congratulations! Your manuscript is now with our production department. 

Kind regards, 

on behalf of

Dr. Pierre Boudinot 

Academic Editor

PLOS ONE